Injuries and molting interference in a trilobite from the Cambrian (Furongian) of South China

Zong Ruiwen zongruiwen@cug.edu.cn
State Key Laboratory of Biogeology and Environmental Geology, China University of Geosciences , Wuhan , China
Wedel Mathew
Electronic publication date: 2021 Apr 7
Publication date: 2021
Volume: 9
Electronic Location ID: e11201
Received 2020 Nov 24; Accepted 2021 Mar 11
Copyright: ©2021 Zong
Copyright year: 2021
Copyright holder: Zong
License: This is an open access article distributed under the terms of the Creative Commons Attribution License, which permits unrestricted use, distribution, reproduction and adaptation in any medium and for any purpose provided that it is properly attributed. For attribution, the original author(s), title, publication source (PeerJ) and either DOI or URL of the article must be cited.
License URL: https://creativecommons.org/licenses/by/4.0/

Keywords: Sub-lethal attack, Shergoldia, Predator-prey interactions, Sandu formation, Guangxi

Funding: National Natural Science Foundation of China 41702006 42072041 Fundamental Research Funds for the Central Universities, China University of Geosciences (Wuhan) G1323520262 This work was supported by the National Natural Science Foundation of China (41702006, 42072041) and the Fundamental Research Funds for the Central Universities, China University of Geosciences (Wuhan) (G1323520262). The funders had no role in study design, data collection and analysis, decision to publish, or preparation of the manuscript.

==============================
An injured Shergoldia laevigata Zhu, Hughes & Peng, 2007 (Trilobita, Asaphida) was collected from the Furongian of Guangxi, South China. The injuries occurred in the left thoracic pleurae possessing two marked V-shaped gaps. It led to substantial transverse shortening of the left pleural segments, with barely perceptible traces of healing. This malformation is interpreted as a sub-lethal attack from an unknown predator. The morphology of injuries and the spatial and temporal distribution of predators indicated that the predatory structure might have been the spines on the ganathobase or ganathobase-like structure of a larger arthropod. There were overlapped segments located in the front of the injuries, and slightly dislocated thoracic segments on the left part of the thorax, suggesting that the trilobite had experienced difficulties during molting. The freshly molted trilobite had dragged forward the old exuvia causing the irregular arrangement of segments. This unusual trilobite specimen indicates that the injuries interfered with molting.

Introduction

Numerous trilobite exoskeleton deformities have been documented, including abnormal healing, hyperplasia, deformation, and missing or fractured segments. The causes of these deformities are usually thought to be injuries, developmental disorders, and diseases (Owen, 1985; Babcock, 1993; Pates et al., 2017; Bicknell & Pates, 2020). The evaluation of injuries caused by predator attack is useful for presenting the interactions between predators and trilobites, and for reconstructing the food web and ecological structure in deep time (Klompmaker et al., 2019). Furthermore, such predatorial injuries are used to uncover behavioral information (Babcock & Robison, 1989; Babcock, 1993; Pates et al., 2017; Bicknell, Paterson & Hopkins, 2019). The injuries caused by predators have mainly been detected on the edges of trilobites, especially in the thoraces and pygidia, and are generally considered to have been non-lethal (Babcock, 2003; Babcock, 2007), while cephalic attacks are more often fatal (Pates & Bicknell, 2019). Although numerous studies have evaluated injured trilobites (e.g., Owen, 1985; Rudkin, 1985; Babcock, 1993; Babcock, 2003; Babcock, 2007; Zhu et al., 2007; Schoenemann, Clarkson & Høyberget, 2017; Bicknell & Paterson, 2018; Bicknell & Pates, 2020; Bicknell & Holland, 2020; Zong, 2020), most predators remain unidentified, except some carnivores with trilobite fragments in their guts or coprolites (Vannier & Chen, 2005; Vannier, 2012; Zacaï, Vannier & Lerosey-Aubril, 2016; Bicknell & Paterson, 2018; Kimmig & Pratt, 2018). The shapes of the Cambrian trilobite injuries suggest that some predators may have been radiodonts (Babcock & Robison, 1989; Babcock, 1993; Nedin, 1999). Other predator candidates include cephalopods, echinoderms, fish, and other larger arthropods (Bruton, 1981; Briggs & Collins, 1988; Babcock, 1993; Fatka, Budil & Grigar, 2015; Jago, García-Bellido & Gehling, 2016; Bicknell & Paterson, 2018; Bicknell et al., 2018a; Zhai et al., 2019).

Moreover, although it has been inferred that injuries did interfere with daily activities of trilobites, there are rare direct fossil records (Šnajdr, 1985). Herein, I discuss an injured Shergoldia laevigata Zhu, Hughes & Peng, 2007 from the Cambrian (Furongian) of Jingxi, Guangxi, South China. The exoskeletal injuries suggest that the predatory structure might have been the spines on the gnathobase or gnathobase-like structure of a larger arthropod. In addition, the findings indicate that these injuries would have caused difficulties for trilobite during molting, but did not cause molting failure.

Materials & Methods

The described Shergoldia laevigata specimen, housed in the State Key Laboratory of Biogeology and Environmental Geology, China University of Geoscience (Wuhan), was discovered from the Cambrian (Furongian)-aged Sandu Formation of Guole Town, Jingxi County, Guangxi Zhuang Autonomous Region, South China (Fig. 1) (Zhu, Hughes & Peng, 2007). The Sandu Formation is represented by calcareous mudstones, siltstones, and argillaceous banded limestones, which formed most probably in the uppermost part of the continental slope (Lerosey-Aubril, Zhu & Ortega-Hernández, 2017). The Sandu Formation is richly fossiliferous, containing abundant, well-preserved articulated trilobites (Han et al., 2000; Zhu, 2005; Zhu, Hughes & Peng, 2007; Zhu, Hughes & Peng, 2010), non-trilobite arthropods (Lerosey-Aubril, Ortega-Hernández & Zhu, 2013; Lerosey-Aubril, Zhu & Ortega-Hernández, 2017), echinoderms (Han & Chen, 2008; Chen & Han, 2013; Zamora et al., 2017; Zamora, Zhu & Lefebvre, 2013; Zhu, Zamora & Lefebvre, 2014), brachiopods, graptolites (Zhan et al., 2010), hyolithids, cnidarians, algae, and some exceptionally preserved soft-bodied fossils (Zhu et al., 2016).

Figure 1 (A) Map of fossil locality at Guole Town, Jingxi County, Guangxi, South China; (B) stratigraphic sketch showing relative position and age of the Sandu Formation.

Trilobites from the Sandu Formation have been classified into at least 25 genera (Zhu, 2005; Zhu, Hughes & Peng, 2007; Zhu, Hughes & Peng, 2010); however, only six abnormal specimens have been documented from this formation (five Tamdaspis jingxiensis Zhu et al., 2007 and one Guangxiaspis guangxiensis Zhou in Zhou et al., 1977; Zhu, 2005; Zhu et al., 2007; Zong, 2020). The injured Shergoldia laevigata was collected from the grey-yellow calcareous mudstones. The specimen is from the Probinacunaspis nasalis–Peichiashania hunanensis Zone of the Furongian, Jiangshanian (Peng, 2009; Zhu et al., 2016).

The fossil in Fig. 2C was whitened with magnesium oxide powder, and all photographs were captured using a Nikon D5100 camera with a Micro-Nikkor 55 mm F3.5 lens.

Figure 2 Malformed trilobite Shergoldia laevigata from the Cambrian Furongian of Jingxi, Guangxi (Specimen No. CUG-GJ-2015-01).

(A) Uncoated specimen; (B) close-up of abnormality in box in (A); (C) specimen whitened by the magnesium oxide powder; (D) sketch of the (A); (E) picture after recovery of the cranidium and the first three thoracic segments, showing the superposed relationship between the posterior area of the fixigena and thoracic segments.

Results

The injured Shergoldia laevigata is preserved as a nearly complete dorsal exoskeleton (30.5 mm long) without librigena, suggesting an exuvia (Daley & Drage, 2016; Drage, 2019). The posterior of the cranidium overlies the first two thoracic segments, this is most pronounced on the left side (Fig. 2). In addition, the first thoracic segment covered most of the left pleural segment of the second thoracic segment, as well as the anterior margin of the right pleural segment. Similarly, most of the first thoracic segment was covered by the posterior area of the fixigena, particularly on its left side. Moreover, the left pleural segments of the fourth to eighth thoracic segments presented an interlaced arrangement, i.e., the anterior margin of the fourth thoracic segment extended upon the third thoracic segment, and the seventh extended upon the sixth (Fig. 2), while there was a typical imbricated arrangement in the right pleural region.

The malformation is on the left part of the exoskeleton, while the medial (axial) and right sections are undamaged. The left thoracic segments are shorter than those on the right side and show limited healing. There are two injuries: one on the third to fourth thoracic segments, and one on the seventh thoracic segment. Two pleural segments are truncated by 3.3 mm because of the first asymmetric V-shaped injury; the most seriously damaged part is the contact site of the two thoracic segments, where there is a V-shaped injury. The second injury truncates the left pleural section of the seventh thoracic segment by 3.5 mm.

Discussion

Possible origin of the injuries and potential predatory structure

Trilobites that are malformed due to predatory attacks have typically V-, U-, or W-shaped injuries (Owen, 1985; Babcock, 1993; Pratt, 1998; Jago & Haines, 2002; Zamora et al., 2011; Pates et al., 2017; Bicknell & Paterson, 2018; Bicknell & Pates, 2020), with a few showing in bay-shaped injuries (Fatka, Budil & Grigar, 2015). Furthermore, there is occasionally signs of healing or regeneration (Rudkin, 1979; McNamara & Tuura, 2011; Pates et al., 2017). In the present specimen, the injuries have traces of healing and are therefore considered evidence of a predatory attack. The two injuries have a similar degree of healing without any regeneration, suggesting that these injuries may have been incurred in the same inter-molt stage.

In the past, the most commonly suggested Cambrian predators are considered to have been the radiodonts, especially anomalocaridids and amplectobeluids, as their frontal appendages and oral cone were extremely effective predatory structures (Whittington & Briggs, 1985; Babcock, 1993; Zamora et al., 2011). Cambrian arthropods or arthropod-like organisms with gnathobases are also considered possible predators, similar to the modern horseshoe crab (Bicknell et al., 2018a; Bicknell et al., 2021). Some amplectobeluid genera have been documented with gnathobase-like structures (Cong et al., 2017; Cong et al., 2018), suggesting that amplectobeluid radiodonts may have been predators of Cambrian trilobites (Bicknell & Pates, 2020). In addition, some trilobites and predatory arthropods with reinforced gnathobasic spines on the protopodal sections of their walking legs are also considered as potential predators (Bruton, 1981, Conway Morris & Jenkins, 1985; Zacaï, Vannier & Lerosey-Aubril, 2016; Bicknell et al., 2018a; Bicknell et al., 2018b; Bicknell, Paterson & Hopkins, 2019; Bicknell & Holland, 2020).

However, so far, the youngest amplectobeluid and anomalocaridid are from the Drumian and Guzhangian, respectively (Lerosey-Aubril et al., 2014; Lerosey-Aubril et al., 2020). Most Furongian and Ordovician radiodonts belong to the family Hurdiidae, which do not have endites of alternating size, and all members of this family are considered to be sediment sifters or suspension feeders (Daley, Budd & Caron, 2013; Daley et al., 2013; Lerosey-Aubril & Pates, 2018; Van Roy, Daley & Briggs, 2015; Pates et al., 2020). Moreover, no radiodonts were discovered in the Sandu Formation. So, the radiodonts are unlikely to have caused the injuries in the Shergoldia laevigata specimen. Gnathobases have a slight size gradation of spines along the gnathal edge (Stein, 2013), or a saw-toothed pattern with spines of alternating sizes (Bicknell et al., 2018a). These spines may caused smaller injuries, that is missing one or two separate thoracic segments, on the edges of trilobites. The arthropods Aglaspella sanduensis (Lerosey-Aubril, Ortega-Hernández & Zhu, 2013) and Glypharthrus trispinicaudatus, Mollisonia-like arthropods, unnamed aglaspidid-like arthropods, Perspicaris-like bivalve arthropods (Zhu et al., 2016; Lerosey-Aubril, Zhu & Ortega-Hernández, 2017), and some larger trilobites (Zhu, 2005) were discovered in the Sandu Formation at the same site. Therefore, the predator who attacked the studied Shergoldia laevigata specimen may be one of these arthropods.

Figure 3 Reconstruction of injury of studied Shergoldia laevigata specimen from the Cambrian Furongian of Jingxi, Guangxi.

(A–B) Predator attack on Shergoldia laevigata, and leading to damage in the exoskeleton. (C) Shergoldia laevigata drag forward the old shell during molting, because of the deformation of the exoskeleton. Such condition leads to the overlap of segments and dislocated arrangement of thoracic segments.

Interference with the molting of trilobite

Previous studies have reported abundant injured trilobites and presented the possible identity of the predators, including information about their behavior (Babcock, 1993; Babcock, 2007; Pates et al., 2017; Bicknell & Paterson, 2018; Bicknell, Paterson & Hopkins, 2019; Pates & Bicknell, 2019). However, there are few direct fossil records showing that injury has disturbed the molting of trilobites (Šnajdr, 1985). The studied specimen has an apparent overlap of segments along with the injuries that are mainly present in the posterior of the cranidium and the front of the thorax, especially in the left part of the exoskeleton. The anterior margin of the injuried third thoracic segment was covered by the unbroken second thoracic segment (Figs. 2A–2D), indicating that the injury formed before the overlap of the segments. Bottom currents can also cause the overlap and even disruption of trilobite segments, the Sandu Formation formed in a relatively calm environment (Lerosey-Aubril, Zhu & Ortega-Hernández, 2017), although there are overlapping segments on the exuvia and carcass of Shergoldia laevigata in the same horizon, their thoracic segments still maintain imbricated arrangement (Zhu, Hughes & Peng, 2007). There is also overlap of thoracic segments on the exuvia of uninjured trilobites (Daley & Drage, 2016), but it is difficult to determine whether it was caused by molting or other abiotic factors. In contrast to the above two cases, in addition to the overlap of segments in the studied specimen, the left thoracic segments are presented an interlaced arrangement rather than imbricated (Fig. 2), which seems not to be caused by bottom currents and is rather likely caused by the active behavior of trilobite. Moreover, all abnormal arrangements of the segments appeared near the injury, and the overlapped part of the segments was only located before the most serious injury (on the third to fourth thoracic segments). It is speculated that all of the irregular patterns were caused by post-injury molting of Shergoldia laevigata. Namely, the new exoskeleton could not be smoothly separated from the old one due to the unbalanced body with injuries (Drage, 2019; Drage et al., 2019). The trilobite dragged forward the old shell to get rid of the exuvia, which led to the overlap of segments and the dislocated arrangement of thoracic segments, especially near the injuries (Fig. 3). Some previous studies have reported cases of failed molting of a trilobite (McNamara & Rudkin, 1984) and other ecdysozoans García-Bellido & Collins, 2004; Drage & Daley, 2016; Yang et al., 2019), in which the new exoskeletons were preserved under the old exuvia. However, none of the fragments of the new exoskeleton were found under or near the exuvia of Shergoldia laevigata, which implies that the molting might not have failed. Although the injuries complicated the molting process, it was successful and the molted trilobite moved away.

Conclusions

The Shergoldia laevigata specimen has substantially shorter pleural segments of the third to fourth and seventh thoracic segments, with signs of lightly healing in the injuries incurred during a sub-lethal predator attack. The degree of healing in both injuries and the distribution of the injuries show that they may have been caused in the same inter-molt stage. Based on the morphology of the injuries and the spatial and temporal distribution of predators, the predatory structure may have been the spines on the gnathobase or gnathobase-like structure of a larger arthropod. The conspicuous overlapping of the segments and dislocated arrangement of the thoracic segments, especially in the left pleural region and near the injuries, shows that the injured S. laevigata encountered certain obstacles during molting. The trilobite dragged the old exuvia forward, which led to the irregular arrangement of the segments. Such configuration can demonstrate that even provisionally healed injury can cause certain complication of the molting process in trilobites.

I appreciate much the constructive and critical comments from Russell Bicknell, Stephen Pates, John Foster and one anonymous reviewer, which aided in the further improvement of the manuscript. I would like to thank Guangchun Zeng (from Guangxi), Qi Liu (from Hunan) and Yonggang Tang (from Shandong) for their help in the field work and collection of specimens.

Additional Information and Declarations

Competing Interests

Author Contributions

Data Availability

The author declares there are no competing interests.

Ruiwen Zong conceived and designed the experiments, performed the experiments, analyzed the data, prepared figures and/or tables, authored or reviewed drafts of the paper, and approved the final draft.

The following information was supplied regarding data availability:

The raw data are photographs in Figs. 1–3.

The specimen is stored in the State Key Laboratory of Biogeology and Environmental Geology, China University of Geosciences, Wuhan, China: CUG-GJ-2015-01.

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
