# Peer review of "Injuries and molting interference in a trilobite from the Cambrian (Furongian) of South China"

_PeerJ, doi:10.7717/peerj.11201_

## Round 0.1 · original submission · Major Revisions

Happily, all four reviewers found the manuscript to be a potentially valuable contribution. I have considered the reviewers' comments and I find them all to be apt and constructive. Please pay special attention to the comments on the identity of the predator, known 'suspect' predators in the same fauna, and the timing of the injuries. I look forward to seeing an improved version of this work in the near future.

·

Basic reporting

Zong presents a new malformed trilobite and discusses possible interpretation of the fossil. I have reviewed this text previously for another journal and find it much improved from the previous version.

However, there are many stylistic errors I have highlighted in the text that need to be addressed before acceptance. Furthermore, the English is in places convoluted and hard to follow, making an interpretation of the material harder. This needs to be improved

There are a selection of citations that are missing and should be considered to help improve the scope and use of the text.

The structure of the text is fine and it is self contained.

Experimental design

The topic is well defined, and the consideration of the fossil and the material is acceptable

Validity of the findings

The findings are valid (this is a single malformed specimen, nothing wrong with that). It is the conclusion that lacks support.

Zong campions the notion that radiodonts, likely Anomalocaris, produced the injury. This is an outdated perspective on Cambrian ecology. Indeed, in the 80s and 90s this view was accepted, but now we know there are groups like Redlichia rex; large trilobites with effective structures for shell crushing. Furthermore, Zong fails to have understood how gnathobasic feeding works. If the author considers that serially arranged appendages with gnathobases were used in feeding (as we see in trilobites and horseshoe crabs) then the injuries—2 V-shaped indentations next to each other—can easily be explained by such feeding. Put differently, more than one leg can be used in mastication and can therefore make multiple injuries on the same side of the carapace. Further to this, there so is no evidence for radiodonts in the deposit, but there are groups that have gnathobasic like structures, like Mollisonia-like arthropods. I think this speaks volumes that the author is supporting the wrong notion here.

I believe that if the author is able to demonstrate the presence of a radiodont in the deposit to support the notion, then there is evidence for their claim. At this point, however; there is, in my perspective, too much evidence for the theory presented by myself and not enough to thoroughly support the view of the author

Reviewer 2 ·

Basic reporting

As the author points out, in vitae damage to paleo specimens is a hot topic at the moment. Food webs in the Cambrian were already complex, with a variety of predators attacking a variety of prey (although I would omit fish line 42. This is the Cambrian). Which predators attacking which prey is the question. The author very nicely reasons out a probable answer to this first question.
Unfortunately the paper falls apart when the author tries to decide if the injury interfered with moulting, a reasonable question that, if answered well, moves knowledge of the Cambrian forward. Some sections are irritatingly repetitive, the same thing (I assume) said in exactly the same words. Had the author recast some of the repetitive statements, I might have been able to discern what the author was attempting to explain. For example, what is meant by the phrase “dragged forward the old exuvia (sic)”? 24, 157, 175
A line editor can insert the plethora of definite and indefinite articles that this paper lacks (at least 3 in the abstract alone). Perhaps the line editor, if feeling magnanimous at the moment, might correct such misuses as most superior (102) and larger (21, 43, 50). But the major flaws in English expression must be addressed by the author, preferably, please, with the guidance of a native English speaker. Examples include 132-134, 151-154, 163-165, which are not clear.

Experimental design

No problem. The analysis of the specimen was good. It could easily be replicated by observing the original specimen.
All the references given in the bibleography are cited in text, and no citations lack a reference.

Validity of the findings

No problem here either

Additional comments

[please send along the comments up in 1, Basic Reporting]

·

Basic reporting

The article is well written in clear English.
On the whole it is well referenced, but I have suggested areas where additional references would allow the author to expand on topics of interest and broaden the discussion.
The article is self contained, and the data presented professionally.

Experimental design

The article presents new data, a single specimen with repaired injuries to one side.

Validity of the findings

The findings are in three parts.
The first, that the specimen represents a trilobite with healed injuries, I agree with.
The second is that the injury was caused by a radiodont, I find more problematic and in need of much more discussion (see detailed comments in next section). At the very least, this should be identified as speculation, as per PeerJ guidelines.
The third, that the injury caused a problem during moulting which the trilobite overcame, I find also problematic and in need of much more discussion to support it. The author notes that this is speculative, however it needs a fuller discussion of alternative causes, and the variation observed in trilobite moulting behaviours, to be sound (see detailed comments in next section).

Additional comments

It is always nice to read a paper considering trilobite malformations, it is a topic I find very interesting. Thank you for working on this project, and I hope you find the points I have raised useful in your revisions.
I have uploaded a .pdf file, with suggestions organised by line number.
These are suggestions of where to expand the discussion through the incorporation of references which may be of interest. It is not necessary to include all the references I suggest in the final manuscript, I include them for completeness.
I also have outlined points you may wish to consider, to make the arguments around the potential predator which caused the injury, and the potential issues with moulting, more comprehensive.

·

Basic reporting

All good, my only suggestions for additional references are in the comments on the MS.

Experimental design

No comment.

Validity of the findings

Overall, looks good, but I had three areas in which, as a reader, I wanted to be more convinced. I've noted some of this in my comments on the MS but some additional points added here:

1) The author raises amplectobeluids among anomalocaridids as possessing among the most predatorially-adapted frontal appendages but says that their structure doesn't match the wounds, preferring Anomalocaris-like frontal appendage morphology as more likely. This might be worth revisiting. I still see Amplectobelua's frontal appendages as compatible and, also considering Daley and Bergstrom's (2012) work regarding anomalocaridid oral cones, I remain skeptical about Anomalocaris or something with a similar frontal appendage being the predator.

2) I think ruling out taphonomic effects more is still important. The break to the 7th thoracic segment looks pretty sharp and jagged (at least in Fig. 2B), compared to that of segments 3 and 4, which seem pretty clearly healed. Also, there is a (compaction?) crack in the left anterior pygidium that appears to continue anteriorly across the 8th thoracic segment and lines up with the break in the 7th. This appears to line up with bends but not breaks in thoracic segments 6 and 5 as well. The possibility that the missing portion of the 7th segment is a taphonomic effect related to that crack should be ruled out, especially given the depth of the lost portion of 7 and the apparent lack of damage to the 6th and 8th segments.

3) Finally, after point 2 is covered but I suppose before getting into 1, I think more should be presented on why the wounds were necessarily caused at the same time. I don't necessarily see a reason why they couldn't have been but I'm not entirely convinced that they were either.

Additional comments

Nice illustrations, clearly written over all.

---

## Round 0.2 · accepted · Accept

Thank you for your diligence in addressing the concerns of the reviewers. I am satisfied with the revised manuscript, and I am happy to accept it for publication in PeerJ.

All three reviewers have further minor suggestions for improvement, which you may wish to address before you upload the final version for publication. You might also consider Reviewer 1's comment on the molting theory, and Reviewer 4's comment regarding the 7th thoracic segment break, and make sure that your arguments and logic are clear.

The decision of whether or not to publish the peer reviews alongside the paper is entirely yours, and will not affect how your paper is handled going forward. However, I encourage you to do so. Making the reviews public allows the reviewers to receive credit for their efforts, and also contributes to the emerging culture of fairness and transparency in editing and peer review.

·

Basic reporting

In reviewing this paper again, I find it much improved linguistically. I have made points in an annotated PDF attached, but overall, it is much easier to follow and I commend the author for putting in this work

Experimental design

More than fine. One specimen, well described. There are a few points, like the use of the word somite, which is rather specific and segment would be better employed. But otherwise fine.

Validity of the findings

The consideration on the molting theory is still a bit convoluted, and the fact that there are two possible explanations makes me wonder which is the more parsimonious of the two. But, I will leave that to the author

Reviewer 2 ·

Basic reporting

Good

Experimental design

Good

Validity of the findings

Good

Additional comments

I am impressed with the quality of the revision and recommend publication. I commend the author on a good job and also a willingness to improve the content and articulation.

There are a few small nits I might pick, all of them mere line edits.
Line 135, ‘maybe’ should be two words, ‘may be’
179, ‘lead’ should be ‘led’; I do not fault the author, because Pb is spelled ‘lead’ which sounds exactly like ‘led’, which…dang, English is confusing.
And there is a lack of definite and indefinite articles toward the end. I trust a line editor will stick some in.
All in all, an excellent job. Thank you!

·

Basic reporting

Fine, no additional comment

Experimental design

Fine, no additional comment

Validity of the findings

See below.

Additional comments

A few edits on the MS including one sentence starting on line 145 that I can't make any sense of -- the first "clause" is itself a complete sentence so I'm not sure how the next three parts are intended to go together.

I'm still not entirely convinced that the 7th thoracic segment break is necessarily an injury and not related to the crack that propagates from the pygidium through T8 and lines up with the T7 break. That the T7 break "looks pretty sharp and jagged" I agree with and is precisely why I worry it is possibly not an injury -- if it showed clear healing like T3-T4 that would be different. But that is for readership to decide and ultimately I may be in the minority in my caution, so this is not a major issue. Over all looks improved.